# The influence of background music and narrative setting on anthropomorphic judgements of killer whale (*Orcinus orca*) emotional states and subsequent donation behavior

Heather M. Manitzas Hill[1]*, Elena Svetieva[2], Sarah Dietrich[3], Emily Gallegos[1], Jeffery Humphries[1], Nicolas Mireles[4], Mario Salgado[1], Roberto Lara[1], Jennifer Zwahr[1]

1 Department of Psychology, St. Mary's University, San Antonio, TX, United States of America,
2 Department of Communication, University of Colorado Colorado Springs, Colorado Springs, CO, United States of America, 3 Department of Communication, University at Buffalo, Buffalo, NY, United States of America, 4 Department of Psychology, The University of Texas at San Antonio, San Antonio, TN, United States of America

* hhil1@stmarytx.edu

**Data Availability Statement:** All data are within the paper and its Supporting Information files.

## Abstract

Animal documentary films such as *Blackfish*, considered nonfiction accounts of reality, nonetheless use rhetorical devices to engage viewers and shape their emotional experience for maximum effect. Such devices can also influence attitudes and alter behavior. In animal documentaries, anthropomorphic impressions of the animals by audiences are key. Using general population samples in the US, three online experiments assessed the influence of background music and narrative setting on how viewers emotionally appraised the emotional state of a killer whale (*Orcinus orca*) and subsequently donated to causes affiliated with killer whales. While happy music led to perceptions of a happy whale, sad music led to perceptions of a sad whale. mediation analyses showed that these perceptions indirectly influence donation behavior, via beliefs about the killer whale's welfare and wellbeing. Analyses also indicated that the highest donation amounts towards killer whales were elicited from footage depicting a killer whale in the wild, with sad background music. These findings highlight the potential power that animal and nature documentaries have over viewers, which, when combined with human tendencies toward anthropomorphism, can have significant influence on conservation attitudes and behavior.

## Introduction

*Blackfish* (2013) was a highly successful film that presented the possibility that killer whales raised in captivity experience extreme emotional stress with fatal outcomes for both the animals and the trainers that interact with them. The film propelled significant legislative change and was credited with single-handedly raising public awareness and influencing public

**Funding:** The St. Mary's University Psychology Department provided partial funding to compensate MTurk participants.

**Competing interests:** The authors have declared that no competing interests exist.

opinion about killer whales in captivity. The powerful impact of this film puts into focus the components and storytelling devices used in mass media, that, when combined with human psychological tendencies towards anthropomorphism, can result in significant social and policy change. Informed by the theoretical, critical, and qualitative work on this topic, the purpose of this research is to experimentally investigate how rhetorical devices in media, such as those used in documentary films about animals, can drive anthropomorphic emotion judgements and impact attitudes and behaviors towards the animals depicted.

## Anthropomorphism

Anthropomorphism is defined as the process by which human attributes, intentions, and motivations are ascribed to nonhuman agents, such as animals or even robots [1]. When humans anthropomorphize animals, they attribute inner states like emotions and cognitions to animal behavior, which can facilitate understanding and promote concern for the well-being of the animal [2].

The Similar Principle Theory states that animals displaying more human-like characteristics are easier to anthropomorphize. Killer whales are animals of which few humans have direct knowledge or experience and they do not express emotion through facial expressions or gestures that are similar to humans. While this makes killer whales harder to anthropomorphize, it also means that storytelling devices in films like *Blackfish* have the potential to have an oversized influence on how the animals depicted are perceived. For example, a March 2014 survey conducted with beach visitors in the Turks and Caicos suggested that while respondents were willing to endorse swim with dolphin programs, they were less likely to endorse marine mammal shows that featured killer whales, with some participants indicating that *Blackfish* had affected their views, although these results were called into question by a subsequent critical review [3, 4].

One of the reasons for this effect may lie in the extent to which storytelling devices in animal film and documentary encourage anthropocentric anthropomorphism. DeWaal [5] distinguished between anthropocentric and animal-centric anthropomorphism as relating to the extent to which humans attribute inner states to animals as a result of our own views and experiences (anthropocentric anthropomorphism) vs attempting to understand their inner states by taking into account the animals' experiences and predilections (animal-centric anthropomorphism). Rowley and Johnson [6] took this distinction to analyze the film *Blackfish* and discussed how it was indeed an example of "anthropomorphic anthropocentrism"—encouraging the attribution of human inner states to nonhuman animals but where human values and experiences are used as the lens through which other life forms can be understood. Rowley and Johnson [6] stated that "the anthropomorphic construction of orcas in *Blackfish* may well be a story that is more about what it means to be human than a story about what it means to be orca" (p. 825). In fact, Robbins et al. [7] found that perceptions of an animal's happiness were strongly driven by evaluations of the way it was living, including the *naturalness* of its life. In the present study, we examine how human values and experiences about freedom (vs captivity) are integral to interpreting and attributing emotional states to animals, a question that has not been addressed systematically. Specifically, we examine how beliefs about whether a killer whale that is filmed in the wild vs captivity influences attributions about its emotional state, and whether it also interacts with other audio-visual narrative framing devices.

## Music as a device that drives emotion anthropomorphism

Music provides emotional tone to footage. The visual content depicted in film and other multi-media provides the object for any music-induced emotion [8–10]. In other words, while

music may produce emotional associations, it is the imagery and narrative in a film that is key in defining the emotion and broadly interpreting the psychological meaning of a situation or making inferences about characters and objects depicted in the film [9–12]. Music-induced emotional associations can drive anthropomorphic processes because these associations lead viewers to access experiences and knowledge from long-term memory and use it as a lens through which to interpret the emotional state of the object, character, or animal depicted on screen [9]. At the same time, the influence of music appears to be largely implicit—when viewers are asked to indicate the influences on their interpretations, they largely attribute it to the visual content, and not the music [13]. Nosal et al. [14] found that pairing ominous-sounding background music with video imagery of sharks resulted in sharks being rated more negatively and less positively as compared to uplifting music or silence. Moreover, participants were more willing to allocate funds toward shark conservation when uplifting music was heard rather than silence while viewing shark imagery. While Nosal et al. [14] established that music affects general attitudes toward animals, in the present study, we hypothesized that the background music of the killer whale footage would influence the anthropomorphic impressions that participants had about the emotional state of the killer whale. Specifically, hearing sad music while watching footage of a killer whale would result in a greater likelihood that viewers would view the whale as experiencing sadness. We further hypothesized that these emotion judgements would be particularly influenced by beliefs about where the killer whale was located (namely, in settings that vary in naturalness: the wild vs captivity), and that viewers will attribute more sadness to killer whales that are believed to be in captivity, regardless of music. Finally, we believed that sad music would interact with the belief that the killer whale is in captivity and would result in more participants endorsing that the killer whale was sad.

## Effects on animal welfare/donation behavior

Importantly, anthropomorphism has been shown to influence a variety of behaviors including conservation efforts, willingness to donate, and actual donation behavior depending on the species and context [15–17]. Many advertising campaigns for the conservation and care of different species capitalize on these biases [16, 17] in crafting and framing their portrayals of animals that will enhance the anthropomorphism process and elicit in viewers an increased willingness to donate. These portrayals often feature the powerful use of narrative, visual framing, and other devices [10, 18, 19]. For example, seeing advertisements of animals in a neglected state or behind bars is correlated with welfare actions and reported distress by humans [20]. For the purposes of our research, we focus on both music and donation target.

**Music.** Although there is no prior research on how anthropomorphic emotion judgements affect perception of animal welfare, there is evidence that music alone can increase donation behavior. Strick et al. [21] experimentally manipulated music that accompanied an advertisement and found that music perceived as "moving" enhanced the amount donated to a cause, regardless of whether the advertisement message had an overall positive or negative tone. The authors found that moving music increases "transportation" into the narrative of the advertisement, providing further support to the idea that music can be used as a device that creates and shapes how viewers interpret visual elements. To the extent that sad music is associated with greater emotional intensity [22], we might expect that the effects of music on donation behavior toward killer whales will be stronger for sad music.

There is little to no empirical evidence, however, that music, even music that creates physiological changes in the form of thrills/chills [23–26], may directly influence donation behavior. Instead, we hypothesize that music sets in motion the attribution of a negative emotional state to another, and this attribution would set in motion empathic processes that culminate in the

perceiver's altruistic motivation toward the target [27]—in this case, the killer whale. When humans believe that a target individual is in distress, they tend to show more helping behavior [28], so it stands to reason that if viewers attribute sadness to a killer whale, the viewers would be more likely to exhibit donation behavior toward killer whale conservation.

**Donation target.** There is some evidence that the target or beneficiary of any donation can influence the likelihood and amount that individuals will donate. Batavia et al. [29], for example, found that conservation outreach messages that highlighted nonhuman animals as beneficiaries were more "successful" (in terms of the likelihood that an individual would donate toward conservation efforts and the amount donated) than those that only highlighted humans as beneficiaries.

Rather than measuring general donation behavior, in our study we wanted to examine how music and setting would influence donation amounts to different *types* of killer whale conservation that each reflected different attitudes toward animal conservation. As such, we measured donations towards a) supporting killer whales in aquariums/captivity, b) supporting killer whales in the wild, and c) supporting efforts to free killer whales from aquariums. To the extent that music interacts with beliefs about the killer whale's setting in order to produce inferences about the cause of the killer whale's emotional state and their well-being, we also expect that donation behavior toward killer whales will be affected. Specifically, we hypothesized that sad music accompanying footage of whales in a purportedly captive setting would enhance anthropomorphic judgements that killer whales in captivity are sad and that they are in a worse state of welfare than killer whales in the ocean, and therefore enhance donations toward "freeing" killer whales from aquariums (and diverting away donations that would support whales in aquariums).

## General materials and methods

### Experimental design

Three experimental studies were administered to US-based MTurk participants who provided informed written consent prior to participating in the study. The studies were conducted according to APA ethical standards and were granted approval by the St. Mary's University IRB committee for human protection. All participants were able to complete only one of the studies. Following several quality control checks (summarized in Fig 1 and below), participants in each experimental study viewed a randomly assigned video that presented a killer whale swimming underwater in an ambiguous background. The same killer whale footage was used

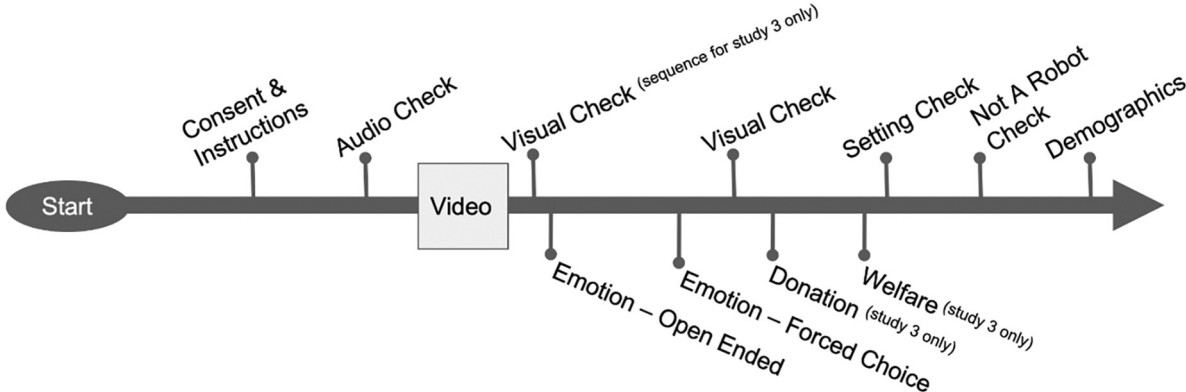

**Fig 1. Timeline of experimental procedures and quality checks.**

for all conditions in this study. In order to assure that the setting was ambiguous, each respondent indicated their perception of the whale's location: 43%-57% indicated the killer whale lived in an ocean, 23%-29% indicated the killer whale lived at a facility, and 20%-28% indicated that the habitat was ambiguous. The variation in responses suggested that the perceived setting of the clip was ambiguous and believable as either a captive or wild setting.

The initial screen information and emotional tone of the background music varied across the experimental conditions. Experiment 1 played the video with one of four background tracks: an aquatic sound (which served as a no-music control condition) or one of three previously established emotion-evoking musical scores: happy, angry, or sad. Experiment 2 manipulated initial screen information to state that the killer whale was in an ocean, in a facility, or a control condition where no setting information was given. Experiment 3 manipulated both the music background and the setting information for a 3 x 3 between-subjects experimental design.

All participants answered the same sequence of questions regarding the video clip they observed (described below), except in Experiment 3 where participants responded to a donation question immediately after the initial emotion questions following the video clip (Fig 1). The survey took approximately 5–10 min to complete, depending on open-ended responses. Participants received $0.10-$0.20 for completing one of the three studies. In the final study, to facilitate donation behavior participants earned an additional $0.50.

## Materials and measures

Three quality control checks were used throughout the study to remove bot responses, inattentive participants, or participants without working audio [best practices, 30, 31]. Fig 1 details the general survey flow used for each experiment. The quality control checks implemented included (1) an audio check before the video in which the participant had to indicate the recorded message, (2) an attention check to ensure the participants had viewed the video, in which the participants were asked to indicate the animal was an orca and not a dog, cat, or sea lion, and (3) at the end of the survey, participants were asked to indicate that a spider did not belong in a grouping of three fruits. Exclusion criteria included the following conditions: participants who failed the quality control checks, surveys that were incomplete for critical dependent variables (i.e., feelings and donations), and surveys that were completed in less time than it would take to watch the video and read the questions. Participants typically completed the survey in three to six minutes. The rate of excluded data varied across the three experiments: Experiment 1–22%; Experiment 2–41%; Experiment 3–32%. The larger percentage of excluded responses in Experiment 2 were composed of 20% that had 0 sec completion time.

Following the video clip, participants responded to a number of questions regarding their perception of the emotional state of the killer whale, first as an open-ended question and then as a forced-choice response. Participants also responded to a variety of questions regarding the perceived welfare of the killer whale, such as how healthy, sick, or cared for the animal appeared, and the purpose of the study. Finally, participants responded to demographic questions, which included questions regarding views of killer whales in captivity and the wild, previous experience with captive facilities, exposure to the movie *Blackfish*, age, gender, education, and geographic location items.

**Experimental video conditions.** The same 19-sec video clip of a killer whale swimming underwater was shown in all experimental conditions (supplemental video files). The 19-sec video was spliced together using two public access videos found on YouTube portraying underwater footage of a killer whale swimming and orienting at an underwater camera (e.g., https://www.youtube.com/watch?v=WTLKNK8K4js). Details of the pilot study and stimulus

selection process may be found in S1 Text. The video clip began with a black screen that said "video loading" or a title screen indicating the setting ("Orca swimming in the Pacific Ocean, "Orca swimming in the marine aquarium, or "Orca swimming underwater"). The video clip title screen lasted for four seconds, faded into the killer whale footage for 15 seconds, and then faded out to a final black screen for two seconds for all experimental conditions presented across all three studies. This formatting standardized the video length and animal view time across conditions with and without introductory text.

**Music manipulation.** For each experimental condition in which the background music was manipulated, three previously validated musical excerpts were selected for their association with a specific emotion [32]. The selection used to evoke the feeling of happiness was *Orchestral Suite No. 2 in B minor, BWV 1068*: *Badinerie* by Johann S. Bach, the selection used to evoke the feeling of anger was *Salome Dances for Peace*: *Half-Wolf Dances Mad* (2004) by Terry Riley, and the selection used to evoke the feeling of sadness was *String Quartet*: *Andante* (2005) composed by Conlon Nancarrow [32]. For the aquatic neutral control condition, the background music was created from a recording of an aquarium pump.

**Coding of free responses.** Free response variables assessed included the emotion the animal was feeling and what was causing the animal to feel that way. Two independent coders (NM and EG) categorized all free responses. Operational definitions for coded categories may be found in Table 1.

Reliability was assessed by two independent raters (JH and MG) for 25% of randomly selected responses to the open-ended question of "How was the animal in the video feeling?" after viewing the video clip for each of the three studies. An overall κ of .69 was found for Experiment 1 between NM and JH. An overall κ of .72 was found for Experiment 2 between NM and JH. An overall κ of .74 was found for Experiment 3 between EG and MG. Table 2 summarizes the individual item reliabilities. HH reviewed and resolved all disagreements for 100% agreement.

**Table 1. Coded categories and operational definitions for open-ended responses.**

| Context | Definition | Exemplars of Participant Responses |
|---|---|---|
| Positive* | A desirable or optimistic form of emotion or expression | "Calm" |
| Negative* | An undesirable or pessimistic form of emotion or expression | "Constrained" |
| Neutral* | Objective form of emotion in which the response was not considered positive or negative | "Neutral" |
| Ambiguous* | A form of expression by a participant that is unclear or that holds multiple interpretations of the subject matter that conflict with one another | "I did not see it feeding." |
| Unsure* | An expression by a participant which demonstrates their uncertainty or lack of confidence on a topic | "Normal?" |
| Setting+ | Any response that hinted towards a Captive, Wild, or Neutral space the animal was in. | "Able to swim freely." |
| Music+ | Any response that had mentioned the musical impact | "The music created the feeling, the animal was enjoying the moment." |
| Human Interaction+ | The participant hints toward a synthetic presence with the animal that alludes to a change in its behavior | "The animal seems to be playing, maybe enjoying the attention from the human camera person." |
| Solitary+ | Any response that states the animal is by itself | "It was all alone. It looked and acted lonely." |
| State, Speed, Movement+ | Any statement that is not considered one of the basic emotions and describes the quality of experience; rate at which the animal was moving; act or state of movement | "To do flips, swimming intensely" |
| Physical Attributes + | Any statement that has to do with a physical characteristic of the animal | "The animal has a smile on its face." |

*Note.* * indicates categories derived for open-ended perceived emotional state of killer.

+ indicates categories derived for open-ended perceived cause of emotional state.

**Table 2. Interrater reliabilities for each coded category per Experiment.**

|  | Experiment 1 | Experiment 2 | Experiment 3 |
| --- | --- | --- | --- |
| Positive emotions | .70 | .79 | .76 |
| Negative emotions | .92 | 1.00 | .77 |
| Unsure | 1.00 | .63 | .74 |
| Ambiguous | .88 | $< .50$ | 1.00 |
| Neutral | $< .50$ | $< .50$ | $< .50$ |

## Experiment 1 –effect of background music on emotional attribution

The purpose of Experiment 1 was to measure the effects of background music on anthropomorphic judgements of a killer whale's emotional state. Specifically, we hypothesized that the background music would be congruent with the judgement such that sad music would enhance anthropomorphic perceptions that the killer whale was sad, and happy music would enhance perceptions that the killer whale was happy).

## Method

### Participants

A total of 359 MTurk respondents participated in Experiment 1. The responses from 280 participants were used following the removal of participants not meeting inclusion criteria (see above in general method section, Fig 2). The modal respondent age was 23–30 years (29%, $n = 81$), followed by 31–40 years (27%, $n = 76$), 41–50 years (19%, $n = 52$), 51–60 years (12%, $n = 33$), 60+ years (9%, $n = 24$), and 18–22 years (5%, $n = 14$). The majority of the sample was female (65%, $n = 182$). The two levels of education with the most respondents were bachelor's degree (39%, $n = 109$) and high school diploma (21%, $n = 60$) with doctoral degrees having the fewest responses (2%, $n = 5$), mirroring educational attainment of the US population [33]. Participants were somewhat evenly distributed across the United States with significantly more respondents reporting from the southeast (32%, $n = 90$) and northeast (20%, $n = 57$) than from other regions.

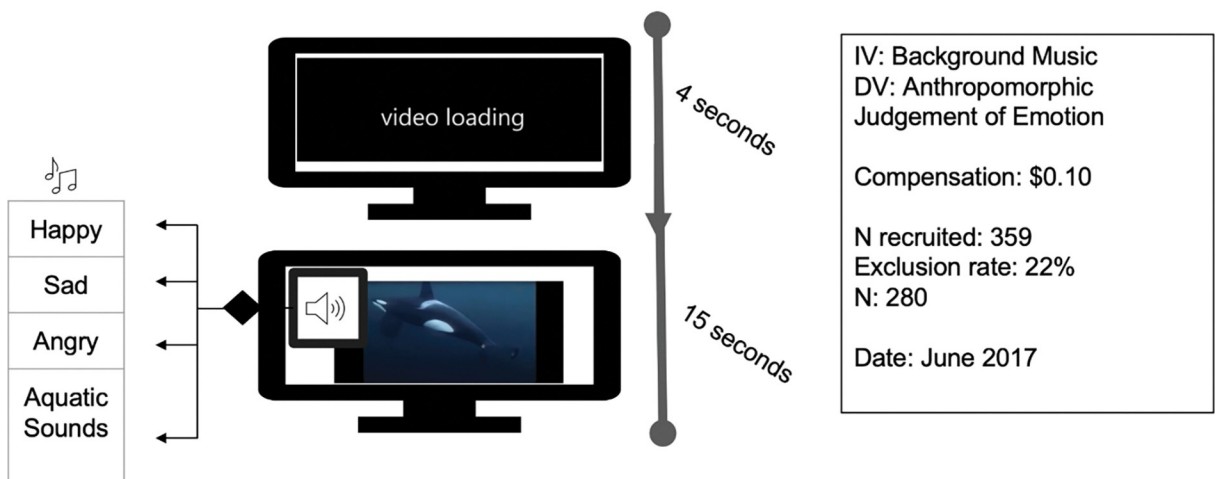

**Fig 2. Experimental design, recruitment, and sampling for Experiment 1.**

## Materials and measures

Four video clips of a killer whale swimming underwater with no visual cues to indicate habitat setting (i.e., wild or captive) were created (see Supplemental files) and randomly assigned following a block procedure. Each video was paired with one of four types of background music: happy, sad, angry, aquatic control (Fig 2). Although angry music had elicited a variety of emotional attributions in the pilot study (see S1 Text), it was included in this experiment to assess the possible cause of the ambiguity—the music or the original killer whale clip. This experiment measured the participants' anthropomorphic judgements of killer whale emotion using both open-ended and forced-choice questions.

## Results

### Relationship between background music and perceived emotion

To investigate how background music influenced judgements of the killer whale's emotion, we conducted a set of binary logistic regressions. The first regression examined how background music influenced the likelihood that an individual would endorse "happy" as the emotion that the killer whale was experiencing. We used planned contrasts for background music where the happy, sad, and angry music conditions were compared with the control/neutral condition (aquatic noises). Compared to the control group, participants who heard happy music while viewing footage of the killer whale were 3.5 times more likely to perceive the killer whale as happy $B = 1.26$ (SE = .23), $W = 30.97$, $p < .001$ ln(B) = 3.51, 95%CI ln(B) [2.26, 5.46]. Participants who heard sad or angry background music were *less* likely to see the killer whale's emotion as happy, but these effects did not reach statistical significance at $\alpha = .05$ level: sad vs control $B = -.44$, SE = .23, $W = 3.70$, $p = .054$ and angry vs control $B = -.38$ (SE = .23) $W = 2.73$, $p = .098$.

A similar pattern emerged when examining the likelihood of appraising the emotional state of the whale as sad. Compared to the control group, participants who heard sad background music were 2.33 times more likely to perceive the whale as being sad $B = .85$ (SE = .29), $W = 8.72$, $p = .003$, ln(B) = 2.33, 95%CI ln(B)[1.33, 4.09]. There appeared to be no effect of hearing happy or angry music on the likelihood of believing that the killer whale was sad, $p > .15$.

### Analysis of open-ended responses

Open-ended coded responses confirmed the validity of the findings based on forced choice. Examples of specific responses and categories may be found in Table 1. The results of a chi-square test of independence indicated a significant association between background music and the subsequent emotion state generated as describing what the killer whale was feeling, $\chi^2(12, N = 281) = 56.15$, $p < .001$, $V = .26$. Significantly more participants rated the killer whale as having a negative emotion after viewing a video clip in which sad music was played (23%, $n = 16$, adjusted residual = 2.8) whereas significantly more participants rated the killer whale as having a positive emotion after viewing a video clip in which happy music was played (68%, $n = 46$, adjusted residual = 5.8). Significantly more participants rated the killer whale as feeling neutral (49%, $n = 35$, adjusted residual = 3.2) after viewing a killer whale video clip with only aquatic sound present.

Viewing a video clip with angry music produced a number of different emotional interpretations, including 11% ($n = 8$, adjusted residual = 3.3) who reported the whale was angry, 28% reported the whale was happy ($n = 20$, adjusted residual = -2.0), and 17% reported the whale was sad ($n = 12$, adjusted residual = 1). Five participants viewed the killer whale video clip with a aquatic control sound, 49% of the participants reported the animal was feeling neutral

($n = 35$, adjusted residual = 3.2), and 28% reported the whale was happy ($n = 20$, adjusted residual = -2.0), 6% reported the whale was feeling sad ($n = 4$, adjusted residual = -2.2).

### Brief discussion

As expected from previous studies, the music heard by participants as they viewed video footage of a killer whale swimming in an ambiguous setting affected their anthropomorphic judgments of the killer whale's emotion. This congruence in music and anthropomorphic judgment of emotions was found for happy music and sad music, and was validated with open-ended responses. As in the pilot study (Supplemental material), the music selected for the angry condition elicited a wide variety of emotional judgments, which also reflects the difficulties researchers face in attempting to use "angry" music (such as rap, heavy metal or Japanese noise music) for the reliable induction of anger [34]. The fact that the findings relating to happy vs sad anthropomorphic emotion judgments as a result of happy vs sad music were consistent across different emotion response formats (forced-choice and open-ended questions), indicates that the effects are reliable and robust, and are not an artifact of response modality.

## Experiment 2—influence of setting beliefs

The purpose of Experiment 2 was to measure the effects of setting on anthropomorphic judgements of a killer whale's emotion. We hypothesized that participant beliefs about whether the whale in the video clip was in the wild or in captivity would also color their judgements of whether the whale was happy or sad. Given widespread anthropocentric beliefs that "freedom" (vs captivity) determines an animal's happiness (vs misery), seeing a killer whale presented as being in a captive setting was expected to lead to judgements that the killer whale is sad while seeing a killer whale represented as being in a wild setting would be more likely to elicit judgements that the whale is happy.

## Method

### Participants

A total of 335 MTurk respondents participated in Experiment 2, and the responses from 196 (40% male) from across the United States were used for this study following the removal of participants not meeting inclusion criteria (see above in general method section, Fig 3). The

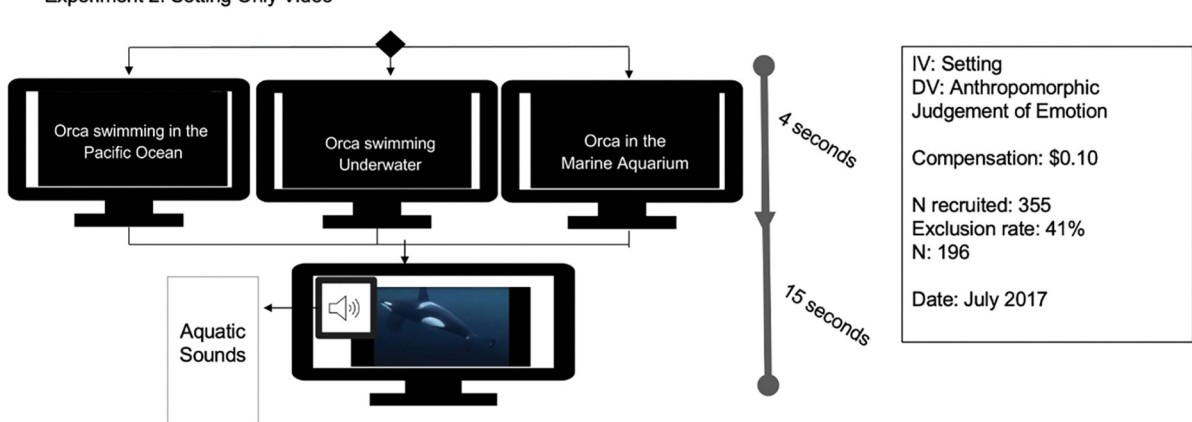

**Fig 3. Experimental design, recruitment, and sampling for Experiment 2.**

modal respondent age was 31–40 (29%, $n = 56$), followed by 23–30 years (26%, $n = 51$), 40–50 years (19%, $n = 37$), 50–60 years (12%, $n = 24$), 60+ years (5%, $n = 10$), and 18–22 years (9%, $n = 18$). The two levels of education with the most respondents were a bachelor's degree (34%, $n = 67$) and a high school diploma (25%, $n = 48$) with doctoral degrees having the fewest responses (3%, $n = 5$). Participants were somewhat evenly distributed across the United States with significantly more respondents reporting from the southeast (30%, $n = 60$) and northeast (19%, $n = 38$) than from other regions.

## Materials and measures

Participants were randomly assigned to one of three setting conditions, killer whale in a captive setting ("orca in the marine water oceanarium"), a wild setting ("orca in the Pacific Ocean"), or neutrally swimming in the water ("orca in the water"). All video clips were played with the same soundtrack of aquatic sounds. Free responses from the survey were coded by raters JH and MG (summarized in Table 1, General Materials and Methods, results in S1 Text). Fig 3 details the experimental manipulation.

## Results

### Manipulation check

Participants were asked to recall the setting in which the killer whale was swimming after identifying that the video clip included a killer whale. The majority of participants (66%, $n = 129$) remembered the setting information accurately: 87% ($n = 41$) correctly identified it as a wild setting, 98% ($n = 41$) correctly identified it as a captive setting, and 75% ($n = 30$) correctly identified it as a neutral setting, Test of Independence $\chi^2(4, N = 129) = 167.30$, $p < .001$, Cramer's $V = .81$. About a third (34%) of the overall sample did not recollect where the killer whale was supposedly living based on the manipulated setting information provided by the video clip they watched. See "setting check" in Fig 1 to see how it fits into the experimental timeline.

### Relationship between setting and perceived emotion

In order to observe how background setting influenced perceptions of the killer whale's emotion, we used a set of binary logistic regressions. The first regression examined how background setting influenced the likelihood that an individual would endorse "happy" as the emotion that the whale is experiencing. We used planned contrasts such that both the wild and captive setting were compared with the control group. Compared to the control group, participants who believed the footage was of a whale in a wild setting were 1.6 times more likely to perceive the whale as happy $B = .48$ (SE = .22), $W = 4.76$, $p = .029 \ln(B) = 1.62$, 95%CI $\ln(B)$ [1.05, 2.45], while participants who believed the footage was of a whale in a captive setting were significantly less likely to believe the whale was happy $B = -.79$ (SE = .26), $W = 9.36$, $p = .002$, $\ln(B) = .45$ 95%CI $\ln(B)$ [.27, .75] compared to the control group.

A similar pattern emerged when examining the likelihood of believing the whale was sad. Compared to the control group, participants who believed the footage was of a whale in captivity were 2.5 times more likely to perceive the whale as being sad $B = .93$ (SE = .30), $W = 9.95$, $p = .002$, $\ln(B) = 2.53$, 95%CI $\ln(B)$[1.42, 4.51]. See Fig 4 for a frequency chart of each emotion across the three setting conditions. Given that only three individuals endorsed "angry" as the whale emotion, we did not conduct analyses on this emotion. We also conducted analyses that controlled for participant gender, but the results did not change from the initial results.

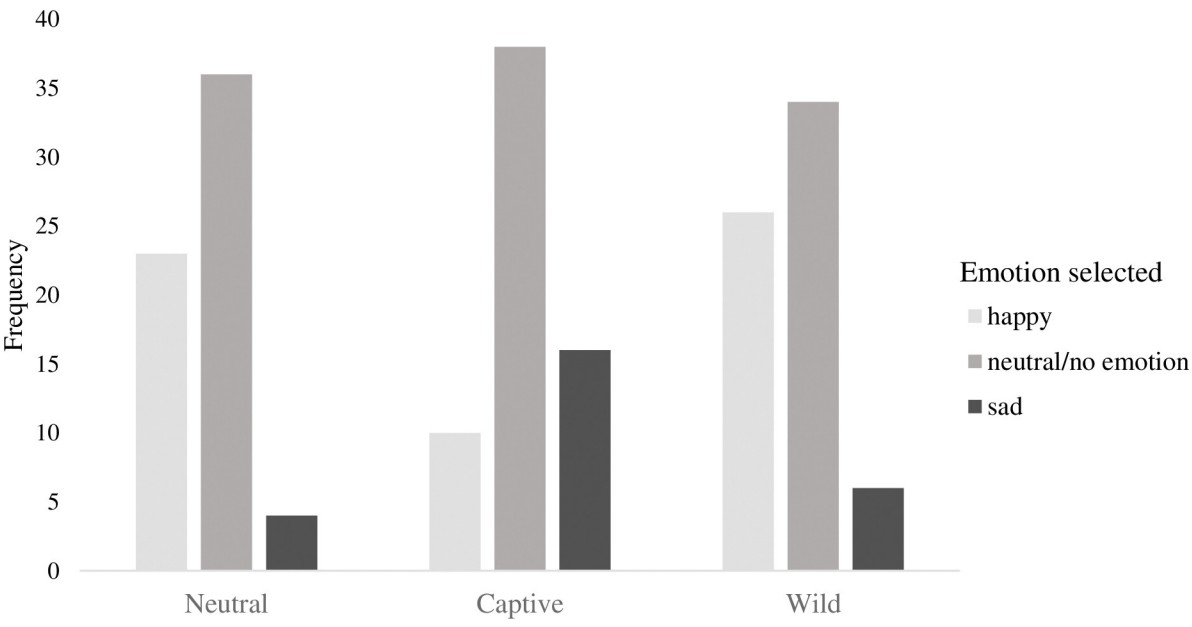

**Fig 4. Frequency of emotion selected across for each experimental setting.**

### Open-ended response

As expected, the results of a chi-square test of independence indicated a significant relationship between background setting and the subsequent open-ended emotion reported by the participant believed the killer whale was experiencing, $\chi^2(8, N = 196) = 23.45$, $p < .003$, Cramer's $V = .25$. Supporting the results of the forced-choice data analysis above, if participants were assigned to a video clip of the killer whale in a captive setting they were more likely to form anthropomorphic impressions of the whale feeling sad (62%, $n = 24$, adjusted residual = 4.2) whereas participants assigned to a video clip that indicated the killer whale was in a wild setting were more likely to form anthropomorphic impressions of the whale feeling happy (41%, $n = 45$, adjusted residual = 2.3). If participants were assigned to a video clip that indicated the killer whale was in a neutral setting the anthropomorphic impressions of the killer whale were equally distributed among the collapsed categories of emotion. Examples of specific responses and their categories may be found in Table 1.

### Brief discussion

As expected, the results of Experiment 2 supported the hypothesis that whales believed to be in a captive setting elicit anthropomorphic impressions of the whale experiencing negative emotion, while a wild setting elicits anthropomorphic impressions of the whale experiencing positive emotion. Ambiguous or neutral setting information produced neutral responses overall. These beliefs were found in both the open-ended, spontaneous responses (see S1 Text results) and in the forced-choice responses, suggesting reliable results.

### Experiment 3—interaction between music and setting information

The purpose of Experiment 3 was to replicate the main effects of Experiment 1 and Experiment 2, as well as to evaluate the presence of interaction effects (including boundary effects) between background music and setting on the anthropomorphic judgements that individuals make about a killer whale's emotional state. Given the results of Experiments 1 and 2, sad music and beliefs

that the footage depicts an animal in captivity both enhance anthropomorphic judgments that the animal is sad. In Experiment 3, we tested the potential additive effects of music and setting beliefs, and hypothesized that sad music would increase anthropomorphic judgments about a killer whale's sadness, but that setting would moderate this effect, such that it would be particularly enhanced when individuals believe they are seeing a killer whale in captivity [25].

The second broad purpose of Experiment 3 was to extend the findings by examining how devices such as music and setting can also influence actual donation behavior toward killer whales, both directly and as mediated via anthropomorphic beliefs about its emotional state. We hypothesized that the emotional quality of the music, combined with beliefs about the killer whale's setting would drive (1) anthropomorphic judgements about the emotional state of the killer whale, (2) perceptions of the animal's welfare, and (3) donations towards animal welfare and the particular type of whale conservation that participants supported. Rather than examine donation behavior as a unitary construct, we aimed to observe how the nature of the donations would change by including the option to donate to (a) free killer whales from aquariums, (b) support killer whales in the wild, or (c) support killer whales living in aquariums.

Given that this is the first study of its kind, we had no clear predictions about how music and setting would influence donation behavior. Nonetheless based on the aggregation of studies across different domains [10, 14, 15, 17, 18], we expected that sad music would drive anthropomorphic beliefs that the whale was sad, and that this, in turn, would drive increased donation behavior toward killer whales in general. We also expected that when participants believe they are watching a killer whale in captivity, anthropocentric beliefs about the importance of freedom would drive increased donations towards freeing killer whales, and that this effect would be especially enhanced when exposed to sad music.

## Experiment 3 method

### Participants

A total of 929 MTurk respondents participated in the study, and the responses from 630 MTurk respondents from across the United States were used for this study following the removal of participants not meeting inclusion criteria (see above in general method section). The modal respondent age was 31–40 (27%, $n$ = 172), followed by 23–30 (27%, $n$ = 170), 40–50 (18%, $n$ = 115), 50–60 (12%, $n$ = 76), 60+ (9%, $n$ = 58), and 18–22 (6%, $n$ = 37). The majority of the sample was female (58% $n$ = 363). The two levels of education with the most respondents were a bachelor's degree (42%, $n$ = 260) and a high school diploma (21%, $n$ = 132) with doctoral degrees having the fewest responses (2%, $n$ = 12). Participants were somewhat evenly distributed across the United States with significantly more respondents reporting from the southeast (26%, $n$ = 163) and midwest (24%, $n$ = 153) than from other regions. The majority of the respondents reported that they disagreed to some degree about keeping killer whales in captivity (81%, $n$ = 509) and agreed that killer whales should be released from captivity (83%, $n$ = 520).

### Design

A 3(Video Settings: wild, captive, or aquatic) x 3(Video Background Music: happy, sad, or neutral) between-subjects design was used for this experiment. The survey was configured so that participants were equally distributed across all the conditions using survey randomization.

### Materials

All participants viewed the same killer whale footage but experienced different title screen information indicating wild, captive, or no information/control setting combined with

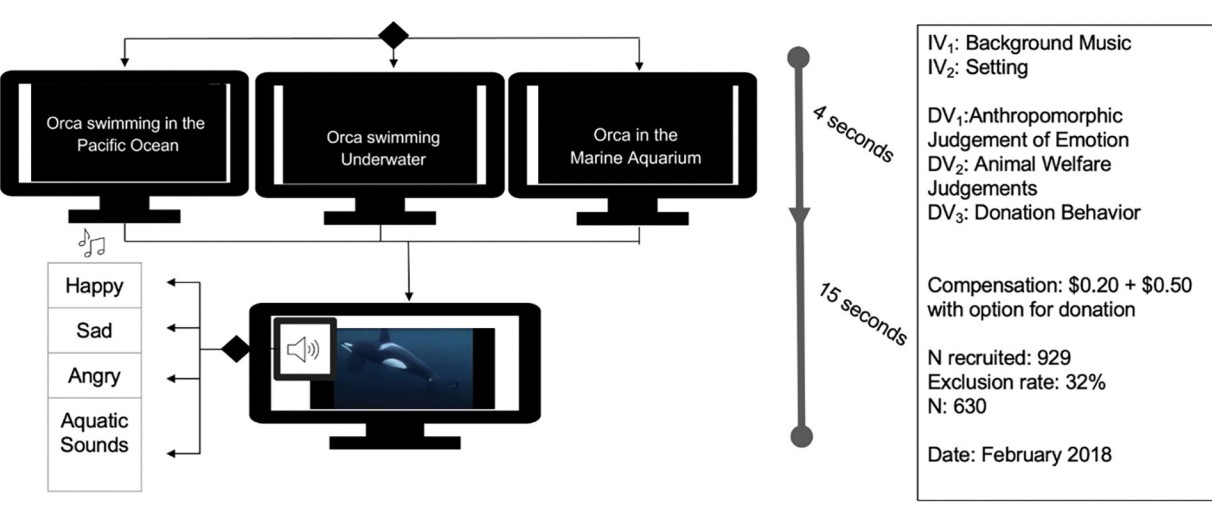

**Fig 5. Experimental design, recruitment, and sampling for Experiment 3.**

different background music: happy, sad, aquatic control. Fig 5 details the experimental manipulation. Following this video clip, participants were asked a series of questions that included: anthropomorphic judgments of the killer whale's emotional state, their confidence that the animal was experiencing that emotion using a 1 to 6 scale (where 6 = *very confident*), and three items measuring perceptions of the animal's welfare (*The animal in the video is sick*, *The animal in the video is well cared for* and *The animal in the video is healthy*, 1 to 6 scale where 6 = *very much agree*), and demographics. Finally, participants had the ability to (re)allocate some of or all of the money they received from participating in the study towards actual donations. Donations could be given to an organization that supports wild orcas, supports orcas in aquariums, or one that frees orcas from aquariums.

## Results

### Anthropomorphic emotion judgments

We utilized the forced-choice response option in addition to the rated confidence variable to derive a single, continuous valenced measure of anthropomorphic emotion judgments. Individuals who rated the whale as experiencing no emotion or "neutral" emotion were coded as "0", while those who perceived the whale as happy or sad were assigned positive or negative values respectively, weighted by their confidence that the whale was experiencing that emotion. Therefore, an individual who felt that the whale was happy, and rated their confidence as 4, was assigned a value of +4, and an individual who believed that the whale was sad, with a confidence of 2, was assigned a value of -2. This variable was used subsequently as the dependent variable in an ANOVA that examined the effects of music and setting on judgements of the whale's emotional state.

**Effects of music.** The analyses revealed a significant main effect of music on anthropomorphic emotion judgements, $F(2,625) = 27.95$, $p < .001$, $\eta_p^2 = .08$ (given the design of these studies, where variance is increased due to the addition of factors, we utilize partial eta square instead of eta squared, allowing a more comparable estimate of effect size for each factor across data collections [35, 36]). A-priori contrasts compared the happy and sad music conditions to the control condition (aquatic sounds). As expected, perceptions of the killer whale's

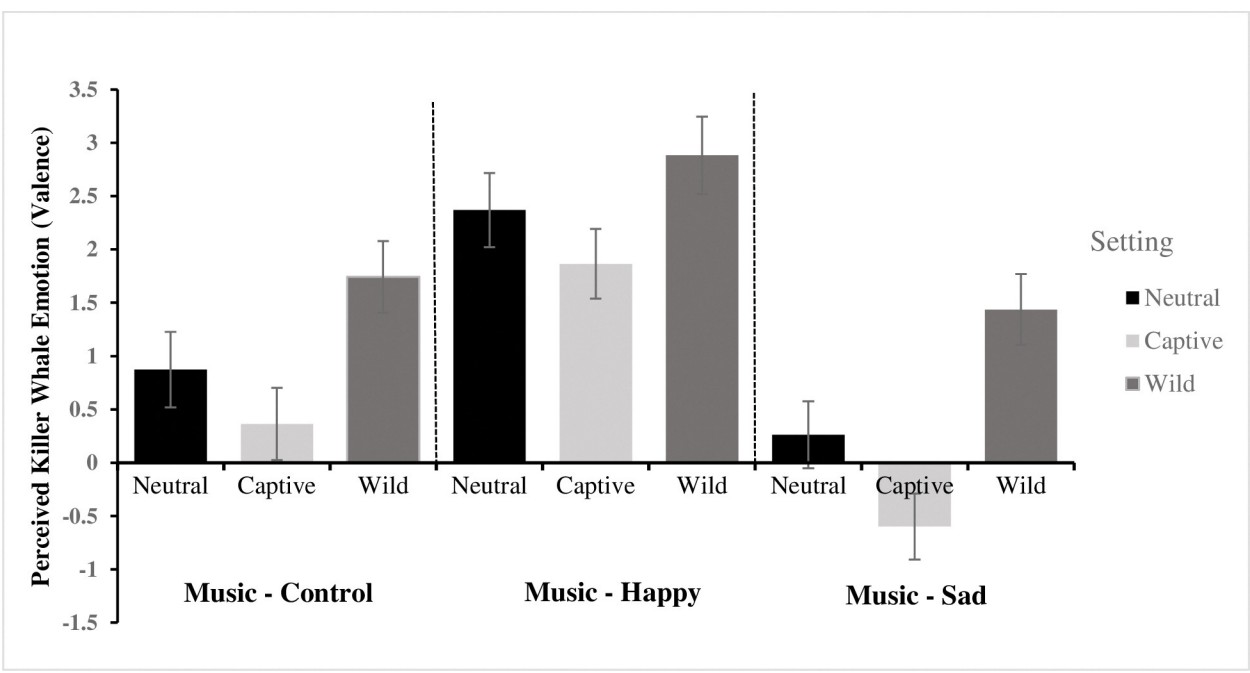

**Fig 6. Perceived killer whale emotion valence across each video setting x video music condition.** Negative values indicate the killer whale's emotion was perceived as being negative. Error bars represent the standard error. *N* = 634; Control/Neutral *n* = 63, Control/Captive *n* = 69, Control/Wild *n* = 70, Happy/Neutral *n* = 65, Happy/Captive *n* = 70, Happy/Wild *n* = 60, Sad/Neutral *n* = 80, Sad/Captive *n* = 82, Sad/Wild *n* = 71.

emotional state were significantly more positive in the happy music condition, contrast estimate = 1.38, *p* < .001 95%CI diff [.83; 1.93], and significantly more negative in the sad music condition, contrast estimate = -.63, *p* = .02, 95%CI diff [-1.16, -.09], replicating the results of the Experiment 1.

**Effects of setting.**   The ANOVA also revealed a significant main effect of setting on anthropomorphic emotion judgements, $F_{(2,625)}$ = 14.65, *p* < .001, $\eta_p^2$ = .05. A-priori contrasts compared the captive and wild setting conditions to the control condition. As expected, perceptions of the whale's emotional state were significantly more positive in the wild setting condition, contrast estimate = .85, *p* = .002 95%CI diff [.31; 1.40], and significantly more negative in the captive setting condition, contrast estimate = -.63, *p* = .022, 95%CI diff [-1.16, -.09], replicating the results of the previous studies.

Analyses indicated no interaction effect between music and setting on anthropomorphic emotion appraisal, *F* < 1. See Fig 6 for means.

## Perceptions of animal welfare

After reverse coding (i.e., scores of a 4 on the item *The animal in the video is sick* were converted to a 1 to be consistent with remaining items), we combined the three animal welfare items into a single measure of animal welfare perceptions (Cronbach's alpha = .87). This variable was used as the dependent variable to test the effects of music and setting on perceptions of the animal's welfare. The ANOVA analysis revealed a significant main effect of both music, $F_{(2, 625)}$ = 6.80, *p* = .001, $\eta_p^2$ = .02, and setting, $F_{(2, 625)}$ = 12.98, *p* < .001, $\eta_p^2$ = .04, on perceptions of the animal's welfare.

Ratings of the animal's welfare were significantly higher for participants who heard happy music (*M* = 4.79, *SD* = .84) compared to both the control condition (*M* = 4.53, *SD* = 0.87); Mdiff = .28 95%CI diff [.07, .48] as well as the sad music condition (*M* = 4.52, *SD* = 0.92);

Mdiff = .27, 95%CI diff [.07, .48]. There was no significant difference between the sad music and control condition. Therefore, the effect of music on perceptions of the animal's welfare was limited to happy music.

Ratings of the animal's welfare were significantly lower for participants who believed they were watching an animal in captivity (*M* = 4.40, *SD* = 0.97) compared to both the control condition (*M* = 4.63, *SD* = 0.85); *M*diff = -.25, 95%CI diff [-.45, -.05] as well as the wild setting condition (*M* = 4.81, *SD* = 0.78) *M*diff = -.43, 95%CI diff [-.63; -.22].

There was no significant interaction effect between music and setting on perceptions of the animal's welfare. These results indicate that merely believing that an animal is in captivity results in decreased perceptions of its welfare, absent any other information. Happy background music to the video footage on the other hand enhances perceptions of the animal's welfare.

## Donation behavior

**Associations with judgements of killer whale emotional state and perceptions of killer whale welfare.**   We examined zero-order correlations between the valenced measure of emotion judgements, perceptions about the animal's welfare, and the amount donated to each donation category (support wild killer whales, support killer whales in aquariums, and free killer whales from aquariums). More positive judgements of the killer whale's emotional state were associated with stronger beliefs that the animal was healthy and well cared for, *r* = .59, *p* < .001, *n* = 635. Positive perceptions of the animal's welfare were generally negatively correlated with donation behavior, and significantly so for donations to free killer whales from aquariums, *r* = -.09, *p* = .027, *n* = 633. That is, beliefs that the animal was not healthy or well taken care of were associated with increased donation behavior towards freeing killer whales from aquariums.

**Direct and indirect effects of anthropomorphic emotion judgements on donation behavior.**   Having established that music and setting influence judgements about the killer whale's emotional state, we wanted to estimate what influence these judgements may exert on donation behavior, both directly, and as mediated via perceptions of the animal's welfare. Mediation analyses were tested using the bootstrapping method with bias-corrected confidence estimates [37–39], with the 95% confidence interval of the indirect effects obtained with 5000 bootstrap resamples [39]. The first step of the mediation confirmed that anthropomorphic judgements of the killer whale's emotional state positively predicted perceptions of its welfare β = .17, *p* < .001. While there was no direct effect of anthropomorphic emotion judgements on the three different types of donation behavior, this does not preclude the possibility of mediation effects [40]. Analyses indicated a significant indirect effect of anthropomorphic emotion judgements on the amount donated to "free killer whales from aquariums", an effect that is mediated by animal welfare perceptions; bootstrap effect estimate = -.02, (SE = .01) 95%CI [-.04; -.002].

**Effects of music and setting on donation behavior.**   In order to estimate the effects of music and setting on general donation behavior toward killer whale conservation, we summed the donation amount across the three types of donation targets: support killer whales in aquariums, support killer whales in the wild, and free killer whales from aquariums (which is equivalent to 100—amount kept for self). A univariate ANOVA, with music and setting as fixed factors and anthropomorphic emotion judgements and perceptions of the animal's welfare as covariates indicated no significant main effect of music, a marginal effect of setting on donation behavior, $F(2,617) = 2.86$, $p = .06$, $\eta_p^2 = .01$, and a significant music x setting interaction effect, $F(4,617) = 2.58$, $p = .04$, $\eta_p^2 = .02$. Perceptions of animal welfare were significantly and negatively associated with donation behavior, B = -7.01, SE = 2.17, *p* = .001 –i.e. more money was donated when individuals had more negative perceptions of the animal's welfare (Fig 7).

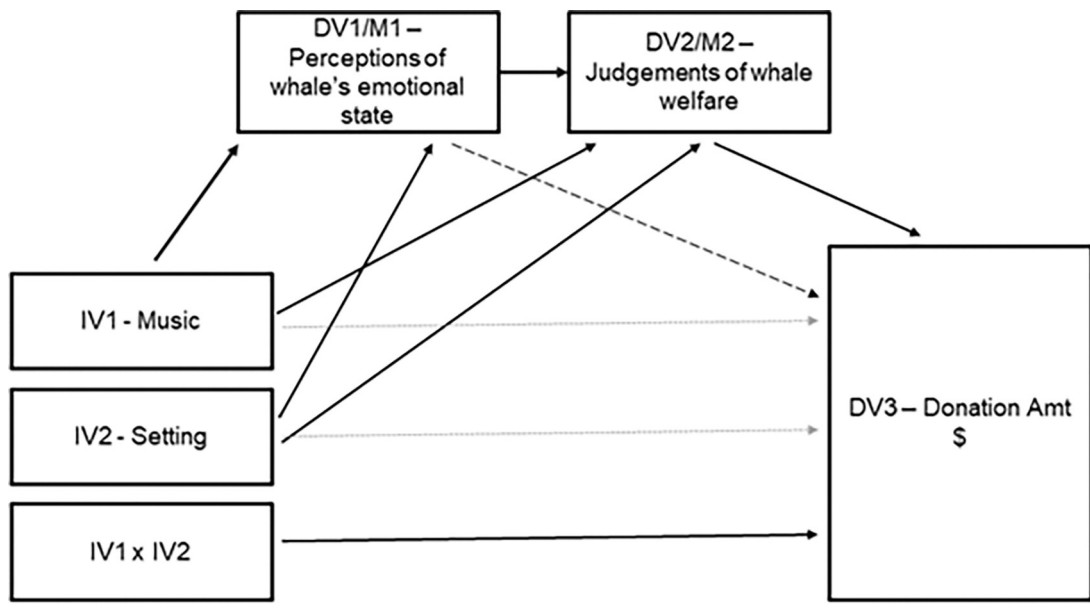

**Fig 7. Mediated path analysis across variables of interest and donation behavior.**

**Simple effects.**   In order to examine the significant music x setting interaction effect more closely, we conducted simple effects analyses. We used ANOVA with experimental condition as the factor, and donation amount as the dependent variable, with post-hoc Dunnett 2-sided contrasts tests between each condition and the control condition (aquatic sounds and neutral setting). This analysis revealed that the condition which featured sad music and presented the whale in a wild setting elicited the greatest donation amounts, mean diff = 19.86, $p$ = .025, 95% CI diff [1.72, 38.01]. Contrary to our hypothesis, sad music while viewing a killer whale that was indicated to be wild (rather than captive) enhanced donation behavior, and particularly to free killer whales from aquariums.

## Experiment 3 discussion

Experiment 3 replicated the results from the first two experiments, with evidence that both background music and purported setting affect the emotional state that individuals anthropomorphically attribute to killer whales. These effects were not additive in that we did not find evidence for an interaction effect. These anthropomorphic emotion judgements had subsequent effects, influencing individuals' beliefs about the animal's welfare, which in turn influences their donation behavior towards killer whales although not in the predicted direction. Our hypothesis was that a sad music background with footage of a purportedly captive killer would enhance donations toward killer whales, but especially for donations to "free killer whales from aquariums". Instead, we found that donations to killer whale conservation were generally enhanced when participants viewed footage of a killer whale that was depicted as being in the wild and was accompanied by sad music.

## General discussion

The purpose of this research was to experimentally investigate how devices used in animal documentary films, such as background music, visual imagery, and narrative framing, can drive anthropomorphic judgements of the emotional state of the animals depicted, and in doing so

impact attitudes and behaviors towards them. To do so, we conducted a series of three experimental studies using US general population samples, where we examined the effect of music, setting information, and then the effects of these two factors combined. Participants were asked to provide judgements of a killer whale's emotional state in all three studies, and in the final study were also invited to donate to different killer whale causes.

## Music's effects on anthropomorphic emotional judgements

Previous research demonstrated that one's interpretation of and reactions to visual imagery and dialogue is influenced by the presence of the background music [10, 14, 19]. While prior research examined emotional reactions to the animals, such as in Nosal et al. [14] where participants provided emotional valence *ratings about sharks* following manipulation of the music soundtrack accompanying the video footage of sharks, the present research supports and extends these findings by showing that music in the context of the visual depiction of animals can affect our anthropomorphic judgements of the emotional state of the animal. Where Nosal et al. [14] found that ominous background music led participants to rate sharks more negatively, and uplifting background music led them to rate sharks more positively, in the current study music affected anthropomorphic judgements about the animal's emotional state. Hearing happy music enhanced perceptions that the killer whale depicted was happy, and hearing sad music enhanced perceptions that the killer whale was sad. Moreover, our forced choice emotion judgements were validated by participants' initial open-ended impressions of the whale's emotional state, providing additional confidence that the finding were not merely a method artifact. Finally, these findings were replicated in Experiment 3 with a different sample and emotion appraisal dependent variable.

Although most research conducted on the effects of music focuses on the reported or physiological responses of *the participants* to different types of music [10, 32], the current study did not measure participants' own emotional responses to the music or the imagery, largely out of a concern that these judgements would color or confound the anthropomorphic emotion judgements of the animal's state. Nonetheless, future experiments may be able to examine this effect, and see whether music affects participants' own emotion states and feelings towards the animals, and whether these changes also drive anthropomorphized impressions of the emotional state of the animals. Moreover, given existing research on the direct effects of mood/emotion on donation behavior [10, 26, 32], this aspect was not a focal research question in our study.

## Emotional judgements and contextualized setting

Experiment 2 assessed how beliefs about the setting of killer whale footage (absent any music) affects participants' judgements about the emotional state of a killer whale. Given existing beliefs about the importance of freedom for animals and that they live in their natural habitats [3, 19], as well as the hyper-focus that films like *Blackfish* place on the purported harm to marine mammals who live in captive facilities, we expected that participants would attribute more sadness to a killer whale they believe to be in captivity than a killer whale in an ocean. Using the same footage as Experiment 1, but pairing it with narrative framing (i.e., setting information), we found that participants were significantly more likely to perceive the whale as happy (compared to a control group) when they believed it was in a wild setting and much more likely to perceive the killer whale as sad when they believed that the killer whale in captivity. These results were replicated in Experiment 3.

Finally we did not find an interaction effect of music and setting on emotion judgements. As seen in Fig 3, the killer whale was perceived generally as having a positive emotional state;

and yet the captive killer whale paired with sad music was the only condition in which the killer whale's emotional state was on average, perceived as negative. While this result failed to reach statistical significance, it is nonetheless in line with our hypotheses. Perhaps the expected interaction would have emerged if participants had been primed for their beliefs about killer whales prior to experiencing the experimental manipulation, rather than as a subsequent demographic question in which the majority endorsed killer whales should be released from captivity; a good avenue for future research. Evidence from a recent experimental study in which participants were more likely to consider a hypothetical chimpanzee experiencing positive emotions and living in an unnatural environment experienced lower welfare than a similar chimpanzee experiencing negative emotions but living in a natural environment [7], supports the idea that human biases influence their perceptions of animal emotional states. The current study did not use actual documentary footage and thus this study is not an ecologically valid test of storytelling devices used in professionally produced visual media. Rather, the current study utilized publicly available footage that was selected for its ambiguity in setting (i.e., participants had difficulty determining the origin of the footage, ocean or aquarium). This decision enabled the study to maintain tight experimental control and provide a relatively *pure* analysis of these effects. The findings should therefore be interpreted as a more conservative test of the power of professionally produced documentaries, which are longer, more engaging, and more vivid.

**Influence of killer whale emotion and welfare perceptions on donation behavior.**
Nosal et al. [14] found a very small effect on donation behavior for sharks, dependent on the background music condition experienced; participants were more likely to allocate funds to protect sharks after hearing uplifting music than a silent condition. We hypothesized that participant beliefs about the emotional state of a killer whale would influence their donation behavior, either directly, or indirectly, via their beliefs about the animal's welfare.

While participants generally kept the majority of their money, the small donations they made to killer whale causes (support wild killer whales, free captive killer whales, or support captive killer whales) nonetheless allowed us a direct behavioral measure of donation. We found that anthropomorphic emotion judgements only indirectly affected donation behavior, via their influence on participant beliefs regarding the welfare of the killer whale in the video footage. First, perceptions of the killer whale's emotion state was correlated with perceptions of their welfare; more positive perceptions of the killer whale's emotional state were associated with stronger beliefs that the animal was healthy and under good care. Second, perceptions of the animal's welfare were in turn negatively correlated with donation behavior, and significantly so for donations to free killer whales from aquariums—beliefs that the animal was unhealthy or not taken care of resulted in higher donation amount to the more radical "free whales from aquariums" cause.

Surprisingly, the experimental condition that elicited the most donation behavior featured a killer whale in the wild that was accompanied by sad music. This finding was contrary to our expectations that the highest donation amounts would flow from seeing a killer whale in captivity, with a sad music background. One possibility is that the combination of sad music with footage of a whale in the wild created a unique aesthetic experience in viewers. Future research will be able to explore individuals' reactions to such footage and determine what changes in emotion (including emotions like awe and elevation, [25]) or cognition/physiology that may have resulted in this particular condition being more effective in eliciting donations compared to the rest. Perhaps participant personal beliefs about killer whales in managed care may have been activated when experiencing this condition, much like participant responses to manipulated vignettes about chimpanzee welfare [7]. Overall, it is clear that music and setting set in motion a number of *unfounded*, anthropomorphic beliefs about the animals' emotional state

and its welfare, which in turn appear to affect donation behavior, specifically donation behavior that is in-line with anthropocentric beliefs about freedom vs captivity.

## Conclusion

This research experimentally demonstrates how common framing devices used in popular visual media, such as films and documentaries, function to activate anthropomorphic cognitive processes and shape how individuals perceive and react to this material. The experiments demonstrate that these effects can be easily observed even with minimal footage, music and narrative framing devices, and measures of subsequent behavior. Using a sample from the general population of the United States, most of whom had not seen the film *Blackfish*, experimental manipulations of music and setting information directly influenced anthropomorphic processes, eliciting congruent emotional appraisals of the killer's emotional state. In simple terms, happy music led to perceptions of a happy whale, while sad music led to perceptions of a sad whale. Killer whales that appeared to be in the wild were perceived as happy, while those in captivity were perceived as sad. Interestingly, the expected additive effect of both the music and the setting did not emerge when the valence of emotional appraisals was examined.

We believe this to be the first study to demonstrate the influence of music and setting on anthropomorphic emotion appraisals. Where Nosal et al. [14] demonstrated that the background music heard effectively altered the positive and negative perceptions participants experienced about sharks, the current study demonstrates that music can also affect the perceptions of how people believe the animal is actually feeling. Similarly, the location of where an animal resides appears to also interact with the existing beliefs of a person and can influence the human perception of how the animal is feeling. Finally, we were interested in determining if these emotional appraisals could influence actual donation behavior as a proxy to understand the effects of visual media on behavior that could potentially facilitate or harm animals whether in human care or their natural habitat. Similar to Nosal and colleagues [14], the effect on donation behavior was subtle and indirect, but more importantly present.

Storytelling devices such as background music and narrative framing, common in both nature film and documentaries, can have a powerful effect on how individuals understand and respond to issues in the natural world, including their willingness to donate to causes and animals deemed to be deserving of help and conservation. Like all such devices, when used irresponsibly or unethically, it can distort understanding and have unintended negative consequences towards bona-fide organizations and conservation efforts. Future research will be able to deepen our understanding of these media devices, and the precise cognitive and emotional responses that shape the people's perceptions and subsequent behavior. Studies such as the current one and that of Nosal et al. [14] highlight the need to recognize that a balanced approach is critical when representing animals in human care or representing animals that are consistently perceived negatively.

## Supporting information

**S1 File. Angry captive video.**
(MP4)

**S2 File. Angry neutral video.**
(MP4)

**S3 File. Angry wild video.**
(MP4)

**S4 File. Aquatic captive video.**
(MP4)

**S5 File. Aquatic neutral video.**
(MP4)

**S6 File. Aquatic wild video.**
(MP4)

**S7 File. Happy captive video.**
(MP4)

**S8 File. Happy neutral video.**
(MP4)

**S9 File. Happy wild video.**
(MP4)

**S10 File. Sad captive video.**
(MP4)

**S11 File. Sad neutral video.**
(MP4)

**S12 File. Sad wild video.**
(MP4)

**S13 File. Raw data from Experiment 1.** Music only.
(XLSX)

**S14 File. Raw data from Experiment 2.** Setting only.
(XLSX)

**S15 File. Raw data from Experiment 3.** Music and setting information.
(XLSX)

**S1 Text.**
(DOCX)

## Author Contributions

**Conceptualization:** Heather M. Manitzas Hill, Sarah Dietrich, Roberto Lara, Jennifer Zwahr.

**Data curation:** Heather M. Manitzas Hill, Elena Svetieva, Sarah Dietrich, Emily Gallegos, Jeffery Humphries, Nicolas Mireles, Mario Salgado, Roberto Lara.

**Formal analysis:** Heather M. Manitzas Hill, Elena Svetieva, Emily Gallegos, Jeffery Humphries, Nicolas Mireles, Roberto Lara.

**Investigation:** Heather M. Manitzas Hill, Sarah Dietrich, Mario Salgado, Roberto Lara.

**Methodology:** Heather M. Manitzas Hill, Sarah Dietrich, Emily Gallegos, Jeffery Humphries, Nicolas Mireles, Mario Salgado, Roberto Lara, Jennifer Zwahr.

**Project administration:** Heather M. Manitzas Hill, Sarah Dietrich, Nicolas Mireles, Mario Salgado.

**Resources:** Heather M. Manitzas Hill, Mario Salgado, Roberto Lara, Jennifer Zwahr.

**Software:** Sarah Dietrich, Nicolas Mireles.

**Supervision:** Heather M. Manitzas Hill, Jennifer Zwahr.

**Validation:** Heather M. Manitzas Hill, Elena Svetieva, Sarah Dietrich, Emily Gallegos, Jeffery Humphries, Nicolas Mireles, Mario Salgado, Roberto Lara.

**Visualization:** Elena Svetieva, Sarah Dietrich.

**Writing – original draft:** Heather M. Manitzas Hill, Sarah Dietrich, Emily Gallegos, Jeffery Humphries, Mario Salgado, Roberto Lara.

**Writing – review & editing:** Heather M. Manitzas Hill, Elena Svetieva, Sarah Dietrich, Nicolas Mireles, Jennifer Zwahr.

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
