## [Decision Letter · Decision Letter 0]

8 Feb 2023

The Influence of Background Music and Narrative Setting on Anthropomorphic Judgements of Killer Whale (Orcinus orca) Emotional States and Subsequent Donation Behavior

PONE-D-23-01403

Dear Dr. Hill,

I was able to get advice from one of the original reviewers of this manuscript. The reviewer felt that the manuscript has improved very much. Therefore, I am happy to accept your manuscript for publication in Plos One.

Kind regards,

José A Hinojosa, Ph.D.

Academic Editor

PLOS ONE

Journal Requirements:

1. Please include your full ethics statement in the ‘Methods’ section of your manuscript file. In your statement, please include the full name of the IRB or ethics committee who approved or waived your study, as well as whether or not you obtained informed written or verbal consent. If consent was waived for your study, please include this information in your statement as well. 

Reviewers' comments:

Reviewer's Responses to Questions

**Comments to the Author**

1. Is the manuscript technically sound, and do the data support the conclusions?

Reviewer #1: Yes

2. Has the statistical analysis been performed appropriately and rigorously? 

Reviewer #1: Yes

3. Have the authors made all data underlying the findings in their manuscript fully available?

Reviewer #1: Yes

4. Is the manuscript presented in an intelligible fashion and written in standard English?

Reviewer #1: Yes

5. Review Comments to the Author

Reviewer #1: This is a much improved manuscript. The text is clear and well written. I feel like I understand the experimental setup better than I did when I read an earlier submission. I recommend that the paper be accepted, but had one minor issue. In lines 373-375, the setting descriptions in quotes do not seem to be word-for-word what appears earlier in the paper or in the little computer schematics in the figures. For example, in the text it says "oceanarium" and elsewhere it says "marine aquarium". I think the authors just need to use the wording exactly as respondents saw it on their screen. So please double check the wording for settings and adjust throughout the ms as needed.

6. PLOS authors have the option to publish the peer review history of their article (what does this mean?). If published, this will include your full peer review and any attached files.

Reviewer #1: No

---

## [Editor Report · Acceptance letter]

28 Apr 2023

PONE-D-23-01403 

The Influence of Background Music and Narrative Setting on Anthropomorphic Judgements of Killer Whale (*Orcinus orca*) Emotional States and Subsequent Donation Behavior 

Dear Dr. Manitzas Hill:

I'm pleased to inform you that your manuscript has been deemed suitable for publication in PLOS ONE. Congratulations! Your manuscript is now with our production department. 

Kind regards, 

on behalf of

Dr. José A Hinojosa 

Academic Editor

PLOS ONE